# Visualising the Emerging Platform of Using Microalgae as a Sustainable Bio-Factory for Healthy Lipid Production through Biocompatible AIE Probes

**DOI:** 10.3390/bios12040208

**Published:** 2022-03-31

**Authors:** AHM Mohsinul Reza, Sharmin Ferdewsi Rakhi, Xiaochen Zhu, Youhong Tang, Jianguang Qin

**Affiliations:** 1College of Science and Engineering, Flinders University, Adelaide, SA 5001, Australia; ahmmohsinul.reza@flinders.edu.au (A.M.R.); rakh0007@flinders.edu.au (S.F.R.); zhu0351@flinders.edu.au (X.Z.); youhong.tang@flinders.edu.au (Y.T.); 2Institute for NanoScale Science and Technology, College of Science and Engineering, Flinders University, Adelaide, SA 5001, Australia

**Keywords:** aggregation-induced emission, visualisation, healthy lipid, green microalgae, health beneficiaries

## Abstract

Nowadays, a particular focus is using microalgae to get high-valued health beneficiary lipids. The precise localisation of the lipid droplets (LDs) and biochemical changes are crucial to portray the lipid production strategy in algae, but it requires an in vivo tool to rapidly visualise LD distribution. As a novel strategy, this study focuses on detecting lipid bioaccumulation in a green microalga, *Chlamydomonas reinhardtii* using the aggregation-induced emission (AIE) based probe, 2-DPAN (C_24_H_18_N_2_O). As the messenger molecule and stress biomarker, hydrogen peroxide (H_2_O_2_) activity was detected in lipid synthesis with the AIE probe, TPE-BO (C_38_H_42_B_2_O_4_). Distinctive LDs labelled with 2-DPAN have elucidated the lipid inducing conditions, where more health beneficiary α-linolenic acid has been produced. TPE-BO labelled H_2_O_2_ have clarified the involvement of H_2_O_2_ during lipid biogenesis. The co-staining procedure with traditional green BODIPY dye and red chlorophyll indicates that 2-DPAN is suitable for multicolour LD imaging. Compared with BODIPY, 2-DPAN was an efficient sample preparation technique without the washing procedure. Thus, 2-DPAN could improve traditional fluorescent probes currently used for lipid imaging. In addition, the rapid, wash-free, multicolour AIE-based in vivo probe in the study of LDs with 2-DPAN could advance the research of lipid production in microalgae.

## 1. Introduction

Lipid droplets (LDs) are well-connected subcellular organelles synthesised through complex pathways in a wide range of organisms. They are involved in vital biological functions, such as cell membrane protection, signalling molecules, cell division, insulation and energy storage. The majority of the lipid compounds have fatty acids (FA) structures, the source of which is either de novo synthesis or exogenous sources [1]. Deregulation in the lipid homeostasis can lead to different pathophysiological conditions in organisms, while specific FAs, such as polyunsaturated fatty acids (PUFAs), are health beneficiaries for higher vertebrates [2]. Therefore, expanding knowledge on lipidomics is a major target for revolution in medical research and system biology [3]. In humans, PUFAs are the precursors of eicosanoids that play pivotal roles in preventing chronic diseases, such as cancers [4] and diabetes [5], through modulating the pro- and anti-inflammatory activities [6]. Long-chain, omega-3 fatty acids such as eicosapentaenoic acid (EPA) and docosahexaenoic acid (DHA) have been linked to the recovery of asthma, cognition, inflammatory bowel disease and brain development [7,8]. Due to the lack of desaturase enzymes, a human cannot synthesise omega-3 fatty acids and, hence, depend on external sources [6]. Plants can de novo synthesise up to 18 carbon PUFAs [9], whereas some fish can convert long-chain PUFA from C18 PUFA [10]. A large portion of the human ω-3 PUFAs is sourced by the fish and a continuous supply of which is currently under threat because of pollution and climate changes [11]. Therefore, alternative sources of PUFAs are critical to explore for human benefits.

Photosynthetic microalgae possess the key desaturases necessary for synthesising long-chain PUFAs [12]. Additionally, with more efficient CO_2_ fixation, these miniature sunlight-driven cell factories can use chlorophyll and other pigments to convert the photon into energy from a harsh environment [13,14]. Therefore, microalgae have been considered to produce health beneficiary bio-functional components over the last decade. However, getting the desired amount and target lipid class from microalgae is still a significant challenge due to growth, lipid quantity, microalgal strains, metabolic and genetic variation [15,16]. Furthermore, lipid composition in microalgae also varies intermittently and can change from saturated to unsaturated lipids in the later phases of lipid accumulation [17]. Therefore, strict temporal and spatial controls on harvesting cells during maximal lipid biogenesis are essential for optimal lipid extraction.

Recently, the intertwined relation between stress response and LD formation in photosynthetic organisms has become apparent [18,19]. Under stress conditions, most microalgae show elevated reactive oxygen species (ROS) activities [20,21]. A high level of oxidative stress can alter photosynthesis and damage the DNA and proteins, leading to autophagy. However, over the last few years, increasing evidence suggests the role of ROS as a mediator of lipid accumulation by acting as the second messenger in various signal transduction mechanisms [22,23]. Compared to other ROS, H_2_O_2_ is much more stable and can cross the phospholipid bilayer to link many complex biological systems [24,25]. Therefore, a controlled surge in H_2_O_2_ production can improve cell functioning and it is necessary to understand the cellular signal transduction mechanisms during lipid bioaccumulation in microalgae [26,27].

Early prediction of the lipid-induced growing phases, the precise LD localisation and identification of the biochemical parameters related to lipid production are critical for designing the lipid induction strategies in microalgae. However, it is necessary to have a reliable, easy and rapid tool in vivo to visualise LD distributions. One of the significant challenges of using traditional fluorophores is developing dye acquisition and penetration techniques. Additionally, commercial fluorescent dyes often suffer from reduced photostability and aggregation-caused quenching (ACQ) phenomena that limit the study efficiency [28,29]. Recent advancement in nanotechnology has introduced several probes in molecular and biological research, which have unique aggregation-induced emission (AIE) properties, an opposite phenomenon of ACQ [30]. These probes show increased emission due to the restrictions of the intramolecular motion through binding with the target biomolecules or during exposure to a specific environment [31].

Recently, AIEgens, 2-DPAN (C_24_H_18_N_2_O) [32] and TPE-BO (C_38_H_42_B_2_O_4_) [33] have been successfully deployed to detect LDs and H_2_O_2_, respectively, in animal cells. The incorporation of AIE-based nanoparticles has improved these AIE probes’ photostability, making these molecules suitable as long-term high contrast imaging tools [34]. However, since these probes have been studied in animal cells, their efficiency needs to be tested in other biological environments with different cellular compositions. Microalgae need to thrive in adverse conditions and they have developed evolutionary conserved complex structures such as cell walls. Besides providing a stable osmotic environment for the cellular machinery, the algal cell wall offers defence against microbial attack and limits the entry of foreign particles. In some cases, these extracellular structures can act as the barrier for the molecular probes that must be overcome to label the intracellular target components effectively. Therefore, efficacies of 2-DPAN and TPE-BO need to be determined across multiple cell types for advanced research in lipid.

In this study, as a model species of microalgae containing carbohydrate cell walls, we used *Chlamydomonas reinhardtii* to identify the lipid inducing conditions with lipid-specific AIEgen, 2-DPAN. We evaluated the effects of light, nutrient starvation, supplemented carbon source on lipid biosynthesis in this microalga. A new H_2_O_2_ detection technique in the lipid-induced *C. reinhardtii* cells was also established with H_2_O_2_-specific AIEgen, TPE-BO. Furthermore, the effect of direct H_2_O_2_ supplementation on lipid production was analysed. More importantly, we have developed a novel method to visualise LDs and H_2_O_2_ in a complex carbohydrate cell wall containing microalga using the lipid and H_2_O_2_-specific AIEgen to understand the LD enriching mechanism in different environments.

## 2. Materials and Methods

### 2.1. Microalga Cultivation and Lipid Induction

Green microalga, *C. reinhardtii* (diameter: 8.0 ± 2.0 µm), was collected from the College of Science and Engineering, Flinders University, Australia. A pure culture of *C. reinhardtii* was established in 500 mL Erlenmeyer flasks containing 250 mL of modified Woods Hole (MBL) medium after autoclaving at 121 °C for 5 min, then cooling at room temperature. The culture medium in 1-L distilled water (DI) contained CaCl_2_.2H_2_O (0.04 g), MgSO_4_.7H_2_O (0.04 g), NaHCO_3_ (0.01 g), K_2_HPO_4_ (0.008 g), NaNO_3_ (0.08 g), EDTA–Na_2_ (0.004 g), FeCl_3_.6H_2_O (0.004 g), CuSO_4_.5H_2_O (0.01 mg), ZnSO_4_.7H_2_O (0.02 g), CoCl_2_.6H_2_O (0.01 mg), MnCl_2_.4H_2_O (0.02 mg), Na_2_MoO_4_.2H_2_O (0.006 mg), C_63_H_88_CoN_14_O_14_P (Vitamin B_12_) (0.0005 mg), C_12_H_17_ClN_4_OS·HCl (Vitamin B_1_) (0.1 mg), C_10_H_16_N_2_O_3_S (Biotin) (0.0005 mg) and C_4_H_11_NO_3_, (Tris(hydroxymethyl)aminomethane) (0.25 g). The pH was adjusted to 7.2 [35]. The culture was then maintained at 25 °C in a temperature-controlled room under continuous light and rotation (70 µmol photons per m^−2^ s^−1^; 100 rpm). All the reagents for growth media preparation were obtained from Thermo Fisher Scientific, Australia. For growth determination, the cells of *C. reinhardtii* were counted on a haemocytometer (Improved Marienfeld Neubauer, Germany).

To find the effects of nutrient and light manipulation on lipid biosynthesis, *C. reinhardtii* cells (0.3 × 10^6^ cells/mL) were taken from the exponential growth phase and subsequently cultured for 8 days in a nutrient-starved stress phase. Five treatments were: (1) MBL medium, (2) MBL without nitrogen (−), (3) MBL without nitrogen (−) and calcium (−), (4) MCM without nitrogen (−) and calcium (−), but with sodium acetate (2.0 g/L) (+) and (5) MCM without nitrogen (−) and calcium (−), but with sodium acetate (2.0 g/L) (+) in the dark. To determine the effects of exogenous H_2_O_2_ application on growth and lipid accumulation in *C. reinhardtii*, cells were cultured in different H_2_O_2_ concentrations (0, 0.4, 0.6, 0.8 and 1.0 mM) supplemented with an MBL medium. Unless specified otherwise, all the experiments were performed in triplicate (*n* = 3).

### 2.2. Study of the Fluorescence Properties of Chlamydomonas reinhardtii and AIEgens

Different natural biomolecules in photosynthetic organisms (e.g., different chlorophyll and accessory pigments) exhibit autofluorescence properties that must be determined to avoid interference during fluorescent staining. The autofluorescence properties of *C. reinhardtii* were detected at different excitation-emission wavelengths using a fluorescence spectrophotometer (Cary Eclipse, MY17180002, Agilent Technologies, CA, USA) (Appendix A). AIEgen, 2-DPAN was obtained from the South China University of Technology, China [32]. The absorption and photoluminescence (PL) spectra of 2-DPAN in DMSO/water mixtures were determined using the fluorescence spectrophotometer with quartz cuvettes of 1 cm path length. Fluorescence spectra of 10 µM 2-DPAN in DMSO (Sigma-Aldrich, St. Louis, MO, USA) and 50–90% water fractions (fw) were measured at room temperature.

### 2.3. Determination of Algal Growth in Different Concentrations of AIEgens and H_2_O_2_

To determine the effects of AIEgens on *C. reinhardtii* growth, cells at the density of 1.0 ± 0.015 (×10^4^) cells / mL were introduced to different concentrations (10, 50 and 100 µM) of 2-DPAN and TPE-BO (Appendix A). The experiment was conducted in an MBL medium and cultured under the previously described condition. The growth rate was determined by counting the cells with a haemocytometer at regular intervals for up to 6 days.

### 2.4. Determination of Hydrogen Peroxide Content

For the colorimetric measurement of H_2_O_2_ content in *C. reinhardtii*, cells cultured in different nutrient altered conditions were harvested by centrifugation for 10 min at 10,000× *g*. The weight of 0.5 g of fresh algae was then homogenised in 5 mL of 0.1% *w/v* trichloroacetic acid (TCA) solution. The supernatant was then collected by centrifugation at 15,880× *g* for 10 min. An aliquot of 0.5 mL of the supernatant was then mixed with 0.5 mL phosphate buffer (10 mM, pH 7.0) and 1 mL potassium iodide (1 M). The absorbance of the solution was measured at 390 nm. The H_2_O_2_ concentration (μmol H_2_O_2_/g fresh weight (FW)) in the sample was then determined using a calibration curve prepared with the known concentrations of H_2_O_2_ [36].

### 2.5. Study of Algal Growth during Hydrogen Peroxide Supplementation

Prior to studying the effects of H_2_O_2_ supplementation on lipid biosynthesis in *C. reinhardtii*, the algal growth at different concentrations of H_2_O_2_ was determined. 1.0 ± 0.015 (×10^4^) cells/mL were introduced to different concentrations (0 mM, 0.4 mM, 0.6 mM, 0.8 mM and 1.0 mM) of H_2_O_2_ and the experiment was conducted in an MBL medium and cultured under the previously described condition. To determine the H_2_O_2_ utilisation by algal cells, 1.0 ± 0.05 (×10^5^) cells/mL were cultured in 0.4 and 0.6 mM H_2_O_2_ supplemented with the MBL medium up to 40 h. H_2_O_2_ concentration in the culture medium was determined colorimetrically as previously described. A control group without algal cells was set against each H_2_O_2_ concentration treatment and the reduction rate of H_2_O_2_ content at each time point in the presence of algae was normalised against the values of the control groups.

### 2.6. Sample Preparation for Fluorescent Staining of Lipid and H_2_O_2_

A stock solution of 1.0 mM BODIPY™ 505/515 in dimethylsulfoxide (DMSO) (Thermo Fisher Scientific Inc., Australia) was prepared and stored at –20 °C in the dark. The stock solution was diluted to 10 μM in DI water and the final concentration of DMSO was adjusted to 0.1% in each sample. Fluorometric quantification of the lipids was done according to the modified method of Cooper et al. [37]. Briefly, the cultured algae were adjusted to 10^6^ cells/mL and the cells were collected by centrifugation at 2000× *g* for 60 sec. Subsequently, the cells were resuspended in 200 μL DI water, gently mixed with 200 μL of the prepared 10 μM BODIPY™ 505/515 solution and incubated in the dark for 5 min. The cells were then washed three times with DI, followed by centrifugation at 2000× *g* for 60 sec and resuspended in water. The stained cells were then protected from light for further analysis.

To determine the fluorescence intensities of different concentration of 2-DPAN (5 µM, 10 µM, 20 µM and 30 µM), *C. reinhardtii* cells (10^6^) of Treatment 4 was used. A 1.0 mM stock solution of 2-DPAN was prepared and was stored at 4 °C in the dark. Fluorescence intensities were determined using the fluorescence spectrophotometer with quartz cuvettes of 1 cm path length. Fluorometric quantification of lipids in *C. reinhardtii* cells with confocal microscopy was done by adjusting the cells at 10^6^ cells/mL. Subsequently, 20 μM 2-DPAN was added, the samples were vortexed at 100 rpm for 30 sec and then stored in the dark for 30 min. The final concentration of the DMSO was adjusted to 0.1% for all the studies. The wash free performance of BODIPY and 2-DPAN was determined by analysing the samples before and after the washing process (wash with DI three times).

TPE-BO was synthesised at La Trobe University, Australia. The fluorescence intensity and H_2_O_2_ specificity of TPE-BO have been previously characterised by Zhang et al. [33]. To determine the H_2_O_2_ activity in *C. reinhardtii*, 1 mM stock solution of probe TPE-BO was prepared in DMSO and was stored at 4 °C in the dark. Algal cells were consequently adjusted to 10^6^ cells/mL and were incubated with 100 µM TPE-BO for 15 min. The final concentration of DMSO was adjusted to 0.1%.

### 2.7. Imaging of Chlamydomonas reinhardtii with Confocal Microscope

The *C. reinhardtii* cells were imaged under a Zeiss LSM 880 Airyscan confocal microscope using ZEN 2.6 software (Carl Zeiss, Australia). The excitation wavelength for BODIPY™ 505/515 and 2-DPAN was 488 nm and the emission wavelengths for BODIPY™ 505/515 and 2-DPAN were 490–517 nm and 526–570 nm, respectively. Excitation and emission wavelength for TPE-BO were set at 405 nm and 428–499 nm, respectively. Auto-fluorescence of the chlorophyll was detected with the excitation wavelength at 488 nm and emission wavelength of 685–758 nm.

### 2.8. Flow Cytometric Analysis of Lipid Content

Cytometric analysis was done using a flow cytometer (model CytoFLEX Flow Cytometer, Beckman Coulter, Inc. USA). Microalgae from different treatments were taken in three replicates and were stained with BODIPY™ 505/515 (5.0 μM) and 2-DPAN (20 μM) according to the previously described methods. FITC (488 nm laser) and KO525-A (405 nm laser) channels were used to detect the fluorescence from BODIPY™ 505/515 and 2-DPAN, respectively. The mean fluorescence intensity from a minimum of 10,000 cells per sample was acquired. Relative fluorescence was determined using the population cell percentile, with cells cultured in the MBL medium serving as the control.

### 2.9. Lipid Extraction and Analysis of Fatty Acids

The total lipid was extracted according to the adapted method of Bligh and Dyer [38]. Briefly, a sample volume was taken from each growing *C. reinhardtii* culture to provide approximately 500 mg of dry algal biomass. The supernatant was discarded after centrifugation at 10,000× *g* and 4 °C for 10 min. The collected pellet was then washed three times with an equal volume of potassium phosphate buffer (pH: 7.2). After that, the pellet was resuspended in DI water and subsequently transferred and weighed in a weighing dish. The samples were dried for 48 h at 60 °C and stored at −20 °C for further analysis. An approximate 50 mg of dried algal biomass was then taken into a pre-washed (with hexane) mortar. Using a Pasteur pipette, the weighing dish was entirely washed with 1 mL of hexane to transfer the samples completely into the mortar. The algal biomass was then grounded into a fine, smooth paste using a pestle. During the grinding process, hexane was added to replenish the evaporated hexane and the resulting slurry was mixed with the pestle until complete homogenisation. All the process was done under the fume hood. After that, the homogenised hexane-cell mass mixture was centrifuged at 10,000× *g* (4 °C, 20 min) and the supernatant was collected in a pre-weighed vial. The pellet was then resuspended with hexane (3 mL) by vigorous vortexing for 1 min and centrifuged at 10,000× *g* (4 °C, 30 min) to ensure the removal of all the cell debris. After complete evaporation of the hexane, the extracted oil mass was then determined gravimetrically. All the lipid extracts were stored at −20 °C for further analysis. Quantitative analysis of fatty acids was done with a Perkin Elmer GC-MS (Clarus 500 and 560S) using internal standards. Crude lipids were extracted from the sample using solvents. The lipids were transmethylated and the profiles of different fatty acids were obtained with gas chromatography. Results were expressed as % of fatty acid methyl esters [39].

### 2.10. Data Analysis

Data from the confocal microscopy were analysed with ImageJ 1.52a [40]. Briefly, the raw images were exported in “tiff” format in ImageJ. Using “Sliding paraboloid”, the background was subtracted from each image and the fluorescence channel was calculated. Subsequently, “Area”, “Mean grey value”, “Integrated density” and “Area fraction” were determined. Fluorescence intensity per cell was then calculated from the total cells by subtracting the background IntDen value from the IntDen value.

The flow cytometry data were analysed using CytExpert v2.4 software and presented as histogram overlays. To detect BODIPY™ 505/515 labelled cells, samples were gated on FITC-A vs. SSC-A. For 2-DPAN, samples were gated on KO525-A vs. SSC-A. Confocal microscopy and flow cytometry data were exported to Excel 2010 to prepare the graphs. The means and medians of different treatments were compared by the relative fluorescence intensity of the gated population, with setting cells from the MBL medium as the control group. SPSS statistics software (version 23) calculated the differences between control and other treatments at the *p* < 0.05 level through the paired sample *t*-test.

## 3. Results

### 3.1. Effects of Nutrient and Light Manipulation on Algal Growth

Except the MCM medium in continuous light, a slower growth rate was found in all treatments (Figure 1). After 8 days of continuous culture, maximum growth of 1.14 ± 0.38 (10^6^) cells/mL was observed in Treatment 1. Nitrogen starvation decreased the algal growth by 16.4% in Treatment 2 than in Treatment 1. A maximum reduction of 57.1% in growth was observed in nitrogen and calcium starved, but sodium acetate was supplemented in the dark condition of Treatment 5. Exposure to light under similar nutrient manipulation increased the algal growth by 26.5% in Treatment 4, while the presence of calcium in nitrogen starved but sodium acetate supplemented culture medium showed the same growth trend in Treatment 3.

### 3.2. Fluorescent Properties of 2-DPAN

The photoluminescence (PL) spectra of 2-DPAN (10 μM) in DMSO was analysed under 405 nm excitation. In the DMSO solution, 2-DPAN showed weak emissions. An absorption peak was observed at 380 nm in a 90% water fraction (Figure 2a). Increasing the water percentage from 50 to 90% also increased the PL intensity of 2-DPAN, which signified the AIE attributes of 2-DPAN (Figure 2b). 2-DPAN was biocompatible compared to the control group and no significant difference was found in the growth pattern of algae even at a very high concentration of 100 µM of 2-DPAN exposure (Appendix A). Following incubation at different concentrations of 2-DPAN (5 µM, 10 µM, 20 µM and 30 µM), an increase in the fluorescence intensities was observed when cells were treated with 20 µM probes compared to that of the 5 µM and 10 µM probes. The differences between the fluorescence intensity of 20 µM and 30 µM were very insignificant (Appendix A). Therefore 20 µM 2-DPAN was used to label the LDs in this experiment. The fluorescence intensity was found to be slightly lower in 10 min incubation than in 30 min incubation (Appendix A). As no significant difference was observed between the 30 min and 60 min incubation, it was assumed that 2-DPAN could completely label lipid drops in *C. reinhardtii* cells within 30 min of the incubation period.

### 3.3. Comparison of the Lipid-Specific Probes

Confocal images of *C. reinhardtii* cells of Treatment 4 (MCM, (−) N_2_, (−) Ca^2+^, (+) sodium acetate (2.0 g/L) (24 h Light) conditions) were used to distinguish the performances of the lipid-specific probes. The lipophilic traditional green fluorescent dye, BODIPY™ 505/515, was used as a control to co-stain the cells [41]. Green (Figure 3c,h) and yellow (Figure 3d,i) fluorescence channels were exclusively assigned to the BODIPY and 2-DPAN, respectively. The merged image indicated that the yellow and green fluorescence channels overlap each other (Figure 3e,j). The arrow position in the merged image (Figure 3e,j) demonstrated almost synchronised intensity changes of 2-DPAN and BODIPY in *C. reinhardtii* cells with a higher intensity for 2-DPAN (Figure 3k). The intensity scatter plot was drafted and the Pearson correlation coefficient and Mander’s overlap coefficient were calculated as 0.91 and 0.92, respectively (Figure 3l). Almost a twofold increase in the fluorescence intensity was observed from the 2-DPAN labelled cells than the BODIPY (Figure 3m). As relative fluorescence per cell indicated the superior performance of 2-DPAN (*p* <0.05) to the BODIPY in the rest of the experiment, the AIE probe, 2-DPAN, was utilised for the confocal analysis of LDs in *C. reinhardtii* cells.

### 3.4. Flow Cytometric Analysis of Lipid Content

The plots from the flow cytometry indicated the efficient labelling of lipid drops in *C. reinhardtii* with BODIPY™ 505/515 and 2-DPAN (Figure 4a–m). Except Treatment 5 (MCM, (−) N_2_, (−) Ca^2+^, (+) sodium acetate (2.0 g/L) (24 h Dark) conditions), cytograms of BODIPY fluorescence vs. side scatter (Figure 4a–e) and 2-DPAN fluorescence vs. side scatter (Figure 4g–k) showed more lipid accumulation in the nutrient-starved cells than that of the nutrient-enriched cells of Treatment 1 (MCM medium under 24 h light condition) (Figure 4a,g). Maximum fluorescence was observed in N_2_ and Ca^2+^ deprived, but sodium acetate (2.0 g/L) supplemented light condition of Treatment 4 (Figure 4d,j), which was followed byTreatment 3 (N_2_ deprived, but sodium acetate (2.0 g/L) supplemented light condition) (Figure 4c,i). Moreover, greater side scatters in the nutrient-starved population indicated the obese phenotypes [42,43] caused by the lipid production and accumulation of probes within the LDs (Figure 4c,d,i,j), which was consistent with the higher relative lipid fluorescence per cell (Figure 4m). As it was evident that the nutrient-starved dark condition of Treatment 5 (Figure 4e,k,m) could not induce lipid production in *C. reinhardtii* cells, this treatment was deduced from the later confocal analysis.

### 3.5. Confocal Analysis of the Stress-Induced Lipid Droplets in Chlamydomonas reinhardtii

Altering the physical and chemical parameters are well-known strategies for lipid induction in microalgae [16,44]. This experiment studied the effects of nitrogen and calcium deprivation, sodium acetate supplementation and light on lipid biosynthesis in *C. reinhardtii*. As the rapid and efficient scanning approach for identifying lipid inducing conditions, the lipid-specific AIE probe, 2-DPAN, was directly introduced to the growing media of different treatment groups (Appendix A). Distinguished LDs from different treatments were clearly visible in the 2-DPAN labelled confocal images (Figure 5). It was evident that Treatment 4 (MBL, (−) N_2_, (−) Ca^2+^, (+) sodium acetate (2.0 g/L)) showed the highest lipid accumulation (Figure 5D,E), followed by those in Treatment 3 (MBL, (−) N_2_, (+) sodium acetate (2.0 g/L), (Figure 5C,E) and Treatment 2 (MBL, (−) N_2_) (Figure 5B,E). The lowest amount of lipid was produced in the nutrient-enriched condition of Treatment 1 (MBL medium) (Figure 5A). Autofluorescence from the chlorophyll was lowest in Treatment 4 (Figure 5D,E), where maximum lipid was produced (Figure 5A,E).

The present study results suggest that 2-DPAN had reasonable lipid specificity and higher sensitivity and penetration abilities to this complex carbohydrate cell wall containing microalgae. It was also evident that without washing, the fluorescence intensity of the BODIPY dye reached outside the meaningful range of the spectrometer (Appendix A), whereas almost no difference in the PL intensities was observed between the wash and wash-free samples with 2-DPAN (Appendix A). Furthermore, confocal images of the wash free BODIPY samples also showed photobleaching of the samples (Appendix A). Therefore, the photostable properties of AIE molecules, wash free and easy sample preparation techniques (Appendix A) could make 2-DPAN superior to traditional BODIPY™ 505/515 dye for rapid visualisation and quick screening of lipid inducing conditions in microalgae.

### 3.6. Hydrogen Peroxide Content in the Chlamydomonas reinhardtii Cells

ROS was associated with the increased metabolite production in microalgae and high levels of H_2_O_2_ inside the microalgal cells were observed in a stress condition [45]. We determined the amount of H_2_O_2_ in lipid-induced *C. reinhardtii* cells. It was apparent that a maximum amount of 38.57 µM H_2_O_2_ was produced in 1 g of fresh microalgal cells of Treatment 4, where the maximum lipid was produced. Compared to the 6.38 µM H_2_O_2_/g of fresh microalgal cells of Treatment 1, about a 3.2-fold and 3.8-fold rise in H_2_O_2_ production was found in the cells of Treatment 2 and Treatment 3, respectively (Figure 6).

### 3.7. Confocal Analysis of H_2_O_2_ Activity in Nutrient-Starved Chlamydomonas reinhardtii Cells

We used an H_2_O_2_-specific AIE probe, TPE-BO, to check H_2_O_2_ in *C. reinhardtii* with different cell membrane structures from an animal cell model [33]. Additionally, labelling LDs with AIE probe 2-DPAN under the same assay condition enabled us to visualise the H_2_O_2_ activities in the cells during lipid induction. From the confocal images, H_2_O_2_ activities were detected in all microalgae cells, whereas more activities were evident in the lipid accumulated (orange arrow) and autophagic cells (white arrow) (Figure 7). It was apparent that with the increase of lipid production in the cells, H_2_O_2_ activities were increased. In comparison to Treatment 1 (Figure 7d,u), a 2.5-and 3.5-fold rise in the H_2_O_2_ production was observed in Treatment 2 (Figure 7i,u) and Treatment 3 (Figure 7n,u), respectively. Maximum 3.8-fold higher H_2_O_2_ was found in Treatment 4, where the maximum lipid was biosynthesised (Figure 7s,u).

### 3.8. Effects of H_2_O_2_ Supplementation on Chlamydomonas reinhardtii

Our analysis showed a clear relation to lipid biosynthesis and intracellular H_2_O_2_ accumulation under nutrient-depleted conditions. As H_2_O_2_ can act as a messenger molecule [26], many complex biological systems have been linked to control the surge of H_2_O_2_ [25,27]. In this study, we studied the effects of direct supplementation of H_2_O_2_ on the growth and lipid accumulation in *C. reinhardtii*. It was apparent that the concentration of H_2_O_2_ is very critical for cells as almost no change in algal growth was observed until 0.6 mM of H_2_O_2_ exposure, while a slight increase in H_2_O_2_ of 0.2 mM from 0.6 mM inhibited the cell growth by 60% (Appendix A). Cells did not survive above 1.0 mM H_2_O_2_ (data are not presented). While monitoring the cells exposed to 0.4 mM (Appendix A) and 0.6 mM (Appendix A) H_2_O_2_, it was obvious that the utilisation of H_2_O_2_ also increased with cell growth, showing a declining trend during the exponential growth period.

### 3.9. Effects of H_2_O_2_ Supplementation on Lipid Bioaccumulation

In comparison to the control (0.0 mM H_2_O_2_) group (Appendix A), cytograms of BODIPY and 2-DPAN labelled cells showed maximum lipid accumulation in 0.6 mM H_2_O_2_ supplementation (Appendix A), followed by the 0.4 mM H_2_O_2_ (Appendix A). Additionally, more obese cells due to the lipid accumulation were also apparent from the more significant side scattering of the H_2_O_2_ supplemented cells (Appendix A). For further clarification, imaging of the 2-DPAN labelled *C. reinhardtii* cells was done with confocal microscopy.

The confocal images also supported the results of the flow cytometry. More LDs were observed in the 2-DPAN labelled H_2_O_2_ supplemented cells (Figure 8), which were almost 5-fold and 4.6-fold higher in 0.6 mM H_2_O_2_ and 0.4 mM H_2_O_2_, respectively, than the control (0.0 mM H_2_O_2_) group (Figure 8m). Interestingly, despite a significantly higher amount of lipid accumulation (*p* < 0.01), there was not any noticeable change in the chlorophyll autofluorescence among the groups (Figure 8m).

### 3.10. Fatty Acid Analysis

Altering the cultural conditions significantly increased the total fatty acid (TFA) content in the *C. reinhardtii* cells. Among the nutrient altered conditions, the maximum amount of TFA (~11% of DW) was found in the cells of Treatment 3 (MBL, (−) N_2_, (−) Ca^2 +^) and Treatment 4 (MBL, (−) N_2_, (−) Ca^2+^, (+) sodium acetate (2.0 g/L)) that was followed by the 8.4% TFA content of Treatment 2 (MBL, (−) N_2_). However, supplementation of H_2_O_2_ in the MBL medium almost increased TFA content by twofold, reaching ~13% in Treatment 1 (MBL medium) (Appendix A). Almost the same trend in the rise of PUFAs was observed in the lipid-induced *C. reinhardtii* cells (Table 1). The PUFAs content was found to be the most at 55.2% in Treatment 4 (MBL, (−) N_2_, (−) Ca^2+^, (+) sodium acetate (2.0 g/L)), which was 17.8% higher than Treatment 1 (MBL medium). Cells of the H_2_O_2_ supplemented MBL medium produced up to ~54% PUFAs, followed by 48.9% PUFAs of Treatment 3 (MBL, (−) N_2_, (−) Ca^2+^). Cells cultured in the nutrient-enriched condition of Treatment 1 produced the maximum amount of SAFAs (~51.6%). Among the PUFAs, α-linolenic acid (ALA) was the most abundant in the lipid-induced conditions of Treatment 3, Treatment 4 and 0.4 and 0.6 mM H_2_O_2_ supplemented MBL medium. In contrast, the nutrient-enriched control condition of Treatment 1 showed the lowest production of ALA. The amount of linoleic acid was almost similar in all the treatments. In all the treatments, the most abundant SAFA was palmitic acid (C16:0), with a maximum production of ~43.7% in Treatment 1.

## 4. Discussion

Although the current levels of lipid research are extremely promising, many research questions are still pending [46]. Many aspects of lipid biosynthesis in microalgae differ among the species due to the evolutionary diversity and the complex regulatory pathways of lipid metabolism [47]. Even significant differences in lipid accumulation have been observed in the same species under the same assay conditions [48]. Therefore, deliberate identification of the lipid-inducing conditions, screening and selecting suitable candidates are the keys to making progress in microalgal lipid research. In this study, using AIE-based techniques, we have identified the lipid inducing conditions in a carbohydrates-based cell wall containing microalgae, *C. reinhardtii*. With lipid-specific AIE probe, 2-DPAN, we determined the impact of different nutrient starvation, carbon source and light on lipid bioaccumulation in this photosynthetic organism. Following the detection of the H_2_O_2_ activities with the H_2_O_2_-specific AIE probe, TPE-BO in the nutrient-starved cells, we further investigated the effects of direct supplementation of H_2_O_2_ on growth, lipid accumulation and fatty acid composition in this microalga.

The results showed the importance of the light, nitrogen and supplemented carbon sources for the growth and lipid biosynthesis in *C. reinhardtii*. Regarding the nutrient-enriched condition, lipid production in the nutrient-starved cells significantly increased, whereas cell growth decreased. The dark condition could not induce lipid synthesis in the nutrient-starved cells (Figure 4). Nitrogen starvation increased lipid production. However, supplementation of sodium acetate as the primary carbon source in nitrogen starved cells showed a marked increase in lipid biosynthesis (Figure 4 and Figure 5). This possibly happened due to the redirection of the metabolic pathway of acetate towards fatty acid biosynthesis instead of acting as the cellular building block precursors through the glyoxylate cycle and gluconeogenesis [49]. All the nitrogen starved conditions showed decreased chlorophyll content (Figure 5 and Figure 7). To adjust the altered cellular metabolism and changed energy content of the cells during nutrient limitation, downregulation of the protein synthesis and photosynthetic apparatus might occur [49,50]. Reduction in the chlorophyll content was, therefore, associated with their utilisation as the internal nitrogen reserve and decreased light uptake rate to minimise excess ROS accumulation [49,51].

While excessive ROS could be lethal for the cells, a certain level can be the secondary messengers in various signal transduction mechanisms, including lipid biosynthesis [52,53]. Among the ROS, H_2_O_2_ can easily pass across the phospholipid bilayer via the aquaporins [24] and, as a key signalling molecule, interact directly with receptor proteins and redox-sensitive transcriptional factors [54,55]. Correlation among H_2_O_2_-induced lipid accumulation, enhanced calcium (Ca^2+^)-ATPase and glutathione activity has been evident in microalgae from previous studies [44,56,57]. Higher lipid production in *C. reinhardtii* cells during N_2_ and Ca^2+^ starvation in this study was possible due to the modulation of H_2_O_2_ and Ca^2+^ pathways, which was also evident from the increased H_2_O_2_ levels in the lipid-induced cells (Figure 6 and Figure 7). H_2_O_2_ perhaps triggered Ca^2+^ signals and transferred the extracellular stimuli to the algal cells to regulate the response mechanisms further. At the point of Ca^2+^ deficiency in the culture media, intracellular Ca^2+^ stores perhaps maintained the cytosolic Ca^2+^ levels [58]. During nutrient starvation, while the synthesis of the membrane components was reduced, photosynthesis continued; hence biosynthesis of the fatty acids was carried on [59]. Generally, photosynthesis generates ATP and NADPH for cell growth and proliferation. While the *C. reinhardtii* cells were starved by the nutrients, growth decreased and NADPH was utilised for FA synthesis [60,61].

To further analyse the effects of H_2_O_2_ on the lipid accumulation and growth of *C. reinhardtii*, we supplemented the nutrient-enriched culture medium with different concentrations of H_2_O_2_. Although exposure of the algal cells to a higher concentration of exogenous H_2_O_2_ (>0.6 mM) significantly decreased the growth (Appendix A), interestingly, the supplementation of 0.4 and 0.6 mM H_2_O_2_ maximised the lipid bioaccumulation without affecting the algal growth and chlorophyll content (Figure 8 and Appendix A). This indicates that the H_2_O_2_ concentration is critical. Therefore, prior to application, there is a need to study the results in different species. Perhaps a specific concentration of H_2_O_2_ (0.4 and 0.6 mM) supplementation induced the cytosolic Ca^2+^ level in *C. reinhardtii* in this study and consequently alleviated the oxidative stress by triggering glutathione (GSH) activity, modulated the calmodulin and *MAPK* expression levels and resulted in increased lipid production [52,56,62]. Simultaneously, improved cellular signalling promoted cytokinesis by persuading the signal transduction cascades without affecting the nucleic acids content, genes expression, cell size and photosynthetic activity that accelerated the growth [63].

We also have reported a novel, rapid and wash-free strategy in vivo to visualise LDs and H_2_O_2_ in *C. reinhardtii* using AIE-based fluorophores. With the increase in water fraction from 50% to 90% in DMSO, the PL intensity of 2-DPAN significantly increased (Figure 2) due to the AIE attributes and the formation of the insoluble nanoaggregates of 2-DPAN in the aqueous media [64]. The successful labelling of LDs with AIEgen, 2-DPAN in *C. reinhardtii* cells was visualised from the confocal images, whereas complete co-localisation with the traditional BODIPY dye was also observed in this study (Figure 3). Therefore, it appeared that penetration of the AIE fluorophore, 2-DPAN, was not hindered by this algal species’ complex carbohydrate cell wall structure. The structure of the lipid-specific AIE probe with negatively charged oxygen atoms attached to the electron acceptor carbonyl group perhaps makes lipid-specific AIE probe hydrophobic [65]. Additionally, the cheap raw material benzophenone to synthesise the keto-salicylaldehyde hydrazine (KSA) derivative, 2-DPAN, is also hydrophobic to increase the hydrophobicity of this AIE probe [32,66]. Being uniquely constituted hydrophobic structure among different organelles, LDs allowed this nano-fluorophore to target and accumulate inside the algal cells strongly. Upon aggregation, KSA formed six-membered ring structures to restrict the intramolecular motion and increase the fluorescence. Additionally, the presence of KSA retained some hydrophilic ability to enhance the probe efficiency in the cells and successfully label the target molecules [32,67,68]. Furthermore, confocal images of the *C. reinhardtii* cells in this study revealed the superior performance of 2-DPAN to the BODIPY (Figure 3), which is possibly due to the presence of the 2-naphthalene group in 2-DPAN to enhance the luminous efficiency of this AIE probe [32]. We also used the AIE probe, TPE-BO, to rapidly detect H_2_O_2_ in *C. reinhardtii* during nutrient manipulation. The simple structure with phenylboronic ester moiety enabled this probe to transform into the phenol group in the presence of H_2_O_2_ [33]. This restricted the intramolecular rotation and turned on the fluorescence, allowing TPE-BO to detect H_2_O_2_ rapidly in this study (Figure 7). Results of this study are also supported by a former study, where TPE-BO detected H_2_O_2_ within seconds in living mice macrophage (RAW264.7) cells and TPE-BO was established as a very low toxic, photostable, highly-effective and very first AIEgen for in vivo H_2_O_2_ research [33].

This study also demonstrated a relatively easy and rapid application method of AIEgen compared to the traditional dye, BODIPY (Appendix A). Using 2-DPAN required fewer steps than the BODIPY, whereas no significant differences in the PL intensities have been observed between the wash and wash-free samples with 2-DPAN (Appendix A). Conversely, BODIPY dye showed unacceptable results and photobleaching in the wash-free samples (Appendix A). Therefore, the dye acquisition techniques with the traditional probes, regardless of BODIPY and Nile Red, require utmost care. Furthermore, using lower concentrations of these commercial probes can cause self-decomposition, while slightly higher concentration or lower-level competencies during washing steps might result in marked background signals. Having the AIE properties, 2-DPAN is free from these limitations. Additionally, 2-DPAN was found biocompatible at much higher concentrations since no negative impact on the growth of *C. reinhardtii* was observed even after exposure to 100 µM of 2-DPAN (Appendix A). Moreover, a previous study suggested that BODIPY could lose 90% of the signals at 2% laser power energy under confocal microscopy within 5 min of irradiation [66]. BODIPY can also unselectively label the mitochondria and nuclear membranes [69].

Besides, the non-fluorogenic nature of BODIPY can produce more background signals, while limited photostability with a slight stokes shift can cause self-absorption of this traditional dye [29]. Another study demonstrated that under 99% laser density, Nile Red could lose more than 95% signal after 20 min and 60 scans without affecting the fluorescent intensity of 2-DPAN [32]. Therefore, these studies indicated excellent photostability of 2-DPAN than the traditional fluorophores. It also appeared that 2-DPAN and TPE-BO did not interfere with the red channel of chlorophyll autofluorescence (Appendix A) or the green channel of BODIPY dye. In addition, a wash-free, rapid and easy experimental procedure allowed in vivo detection of LDs and H_2_O_2_ under the same assay condition. Thus, this study signifies the effectiveness of these AIEgens as multicolour visualising tools for the target molecules in microalgae.

Moreover, FAs can affect human health and well-being through their direct and indirect association with different cellular functions. The effects of saturated fatty acids (SAFAs), monounsaturated fatty acids (MUFAs) and PUFAs are distinct. Thus, the FAs classes influence various diseases, including cardiovascular disease, metabolic disorder, type 2 diabetes, inflammatory diseases and cancer [70,71]. In this study, the FA profile of *C. reinhardtii* consisted of a range of SAFAs, MUFAs and PUFAs, whereas palmitic acid, linoleic acid and α-linolenic acids were predominant (Table 1). As *Chlamydomonas* and higher plants retained the same ancestry [72], the FA composition of *Chlamydomonas* is almost similar to that of higher plants like *Arabidopsis thaliana* [73]. However, cultural conditions can significantly affect FAs content and composition [74], which is also evident in this experiment as lipid-induced conditions of Treatment 3 (MBL, (−) N_2_, (−) Ca^2+^), Treatment 4 (MBL, (−) N_2_, (−) Ca^2+^, (+) sodium acetate (2.0 g/L)), 0.4 and 0.6 mM H_2_O_2_ supplemented MBL medium produced more ALA than the control group (MBL medium).

As an essential fatty acid, the beneficiary effects of ALA have been suggested by former cohort studies since it could have anti-inflammatory [75,76], anti-arrhythmic [77], anti-thrombotic [77,78] and neuroprotective effects [79] in humans. Other prospective studies suggested improved effects of ALA intake on fatal coronary heart disease (CHD) and gene programming for preventing metabolic diseases [80,81,82]. ALA is easier to burn by the body than palmitic, stearic, oleic, or linoleic acid and, therefore, is expected to induce fat utilisation as energy and reduce obesity [83]. In a study on adult males and females, ingestion of 2.6 g/day ALA reduced systolic and diastolic blood pressure without any negative impact on oxidation or blood coagulation [84]. Moreover, ALA can also be converted into eicosapentaenoic acid (EPA) and docosahexaenoic acid (DHA) in humans. EPA can be utilised to improve conditions like asthma and atopic diseases. As the building block precursors of prostaglandins, leukotrienes, thromboxanes and lipoxins, EPA also plays important roles in different bio-physiological processes [85,86]. DHA can support cell membrane and signalling, specifically in the brain and retina, allowing improved visual activities, immunity and cognitive functions in humans [87,88]. However, the conversion occurs to a limited extent, lower for DHA than EPA and is higher in women than men [89,90]. As the lipid inducing conditions of Treatment 3 (MBL, (−) N_2_, (−) Ca^2+^), Treatment 4 (MBL, (−) N_2_, (−) Ca^2+^, (+) sodium acetate (2.0 g/L)), 0.4 and 0.6 mM H_2_O_2_ supplemented MBL medium in this study produced more ALA. The results show more promises to increase the availability of this bio-functional component in this green microalga *C. reinhardtii*.

## 5. Conclusions

This study identified N_2_ and Ca^2+^ starvation; N_2_ and Ca^2+^ starvation, but sodium acetate (2.0 g/L) supplementation; 0.4 and 0.6 mM H_2_O_2_ supplementation in MBL medium under 24 h light as the health beneficiary lipid-inducing conditions in *C. reinhardtii*. We also detected the H_2_O_2_ activities in the cells and determined that a certain amount of H_2_O_2_ supplementation could effectively induce the production of bio-functional components in microalgae. We have introduced a rapid, wash-free, multicolour imaging technique to visualise LDs and H_2_O_2_ in *C. reinhardtii* with AIE-based bioprobes. These AIE probes are cheap, biocompatible and convenient for high contrast visualisation of LDs and H_2_O_2_ in in situ conditions. Due to the AIE properties, 2-DPAN surpassed the performances of the traditional fluorophore, BODIPY. The AIE probes were able to efficiently localise the LDs and H_2_O_2_ by penetrating the complex carbohydrate-based cell wall of *C. reinhardtii*. As reliable and easy bioimaging tools, these probes are more auspicious to identify the optimum lipid production conditions at early cultural stages in microalgae, which would be more cost-effective, timesaving and labour-efficient. In this study, we have also reported cultural conditions that can produce more health beneficiary α-linolenic acid, which is important for preventing life-threatening diseases in humans. This new technique will significantly ease microalgal lipid research. However, more studies focusing on other cell types, such as diatoms with silica cell walls, are recommended for further assurance of the reliability of these probes for all kinds of microalgae. Thus, it will broaden the horizon of these probes for biological research.

## Figures and Tables

**Figure 1 biosensors-12-00208-f001:**
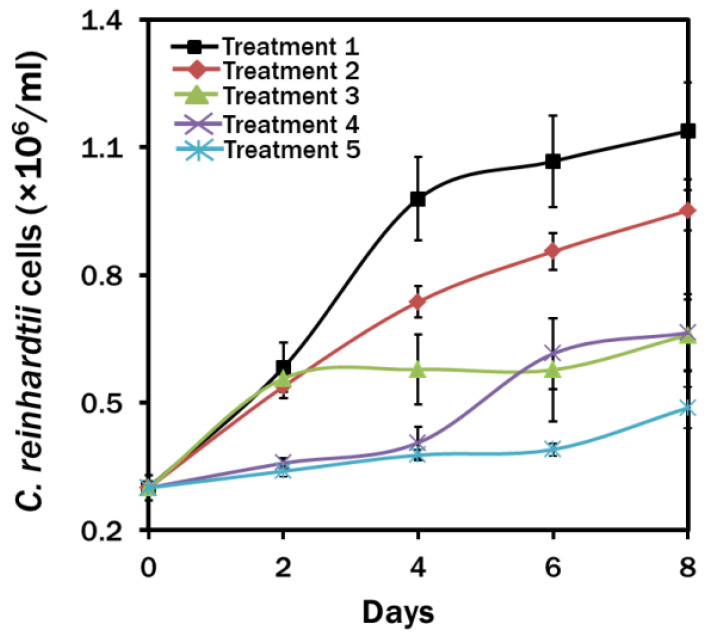
Effects of nutrient and light manipulation on the growth of *Chlamydomonas reinhardtii* at different time intervals. Treatment 1: MBL medium; Treatment 2: MBL, (−) N_2_; Treatment 3: MBL, (−) N_2_, (−) Ca^2+^; Treatment 4: MBL, (−) N_2_, (−) Ca^2 +^, (+) Sodium acetate (2.0 g/L) (24 h light) and Treatment 5: MBL, (−) N_2_, (−) Ca^2+^, (+) Sodium acetate (2.0 g/L) (24 h dark) conditions.

**Figure 2 biosensors-12-00208-f002:**
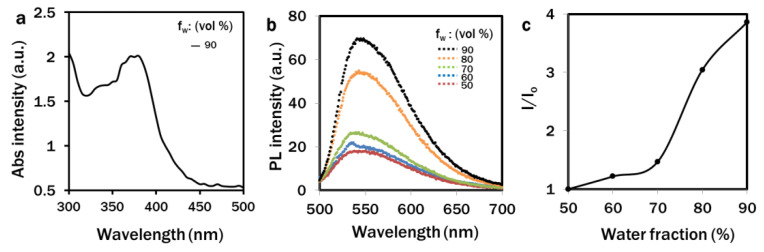
Fluorescence properties of 2-DPAN. Absorption spectra (**a**) and fluorescence spectra (**b**,**c**) of 2-DPAN (10 µM) in 90% water fraction and DMSO-water mixtures, respectively.

**Figure 3 biosensors-12-00208-f003:**
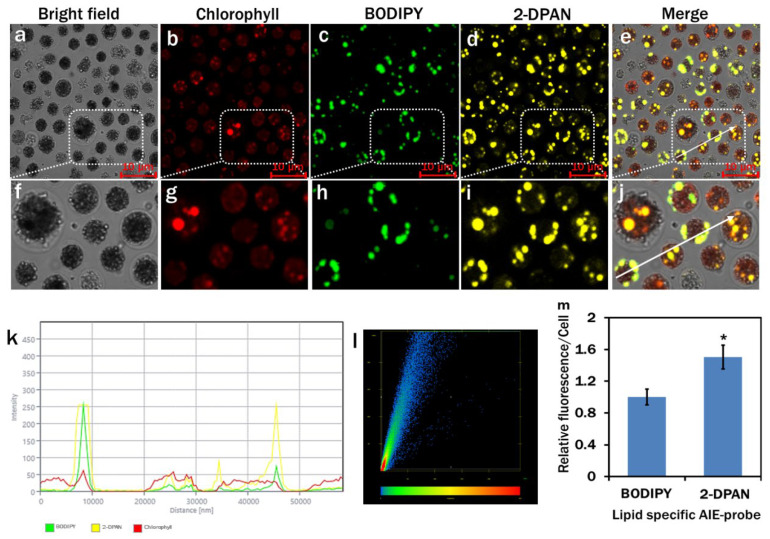
Confocal images of lipid drops in *Chlamydomonas reinhardtii* cells of Treatment 4 (MBL medium, (−) N_2_, (+) Sodium Acetate (2.0 g/L), (−) Ca^2+^, (24 h light condition)). Cells were labelled with traditional BODIPY™ 505/515 (5 μM; incubation-5 min) (**c**) and AIE probe, 2-DPAN (20 μM; incubation-30 min) (**d**). (Bright-field images: (**a**); Fluorescence images—Chlorophyll: (**b**) (λ_ex_: 488 nm, λ_em_: 685–758 nm); BODIPY: (**c**) (λ_ex_: 488 nm, λ_em_: 490–517 nm); 2-DPAN: (**d**) (λ_ex_: 488 nm, λ_em_: 570–650 nm); Merged image: (**e**–**i**) and (**j**) are the enlarged regions of a–d respectively); (**k**) intensity profile of BODIPY and 2-DPAN in green and yellow channels, respectively; (**l**) intensity scatter plot for the colocalised channels; Pearson correlation coefficient and Mander’s overlap coefficient were calculated as 0.91 and 0.92, respectively; (**m**) Relative fluorescence intensity/cell for lipid-specific probes. Values are relative to the control BODIPY dye. Averages shown as mean ±SE; * *p* < 0.05. Images were taken with Zeiss LSM 880 Airyscan confocal microscope.

**Figure 4 biosensors-12-00208-f004:**
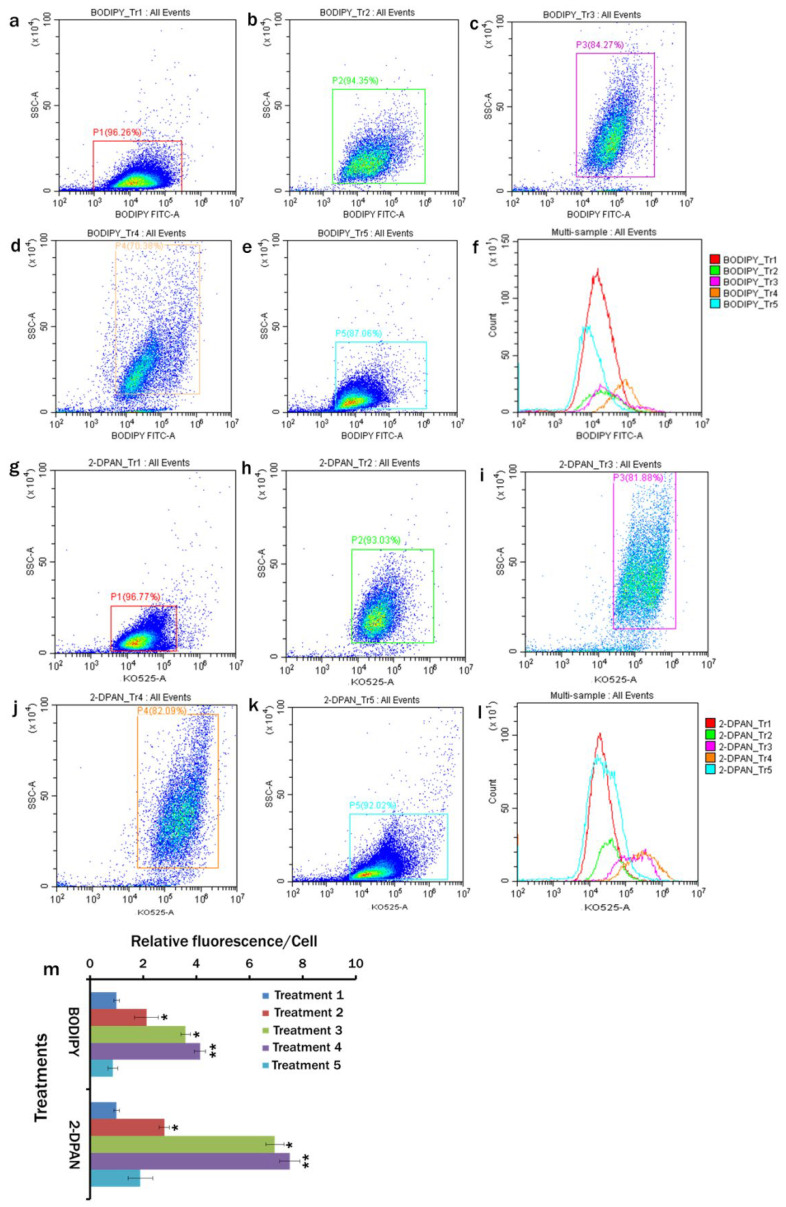
Flow cytometry measurements for lipid in *Chlamydomonas reinhardtii* cells in different treatments labelled with BODIPY™ 505/515 and AIE probe, 2-DPAN. (**a**–**e**) Flow cytogram of FITC-A vs. SSC-A for BODIPY fluorescence in different treatments; (**g**–**k**) Flow cytogram of KO525-A vs. SSC-A for 2-DPAN fluorescence. Cells were cultured in Treatment 1: modified Woods Hole (MBL) medium; Treatment 2: MBL, (−) N_2_; Treatment 3: MBL, (−) N_2_, (−) Ca^2+^; Treatment 4: MBL, (−) N_2_, (−) Ca^2+^, (+) sodium acetate (2.0 g/L) (24 h light); and Treatment 5: MCM, (−) N_2_, (−) Ca^2+^, (+) sodium acetate (2.0 g/L) (24 h Dark) conditions. Histogram of BODIPY™ 505/515 (**f**) and 2-DPAN (**l**) fluorescence for cells. (**m**) Relative fluorescence of BODIPY™ 505/515 and 2-DPAN / cell for different treatments. Values are relative to the control condition of Treatment 1. Averages shown as mean ± SE; * *p* < 0.05; ** *p* < 0.01. All plots are in the logarithmic scale for both axes.

**Figure 5 biosensors-12-00208-f005:**
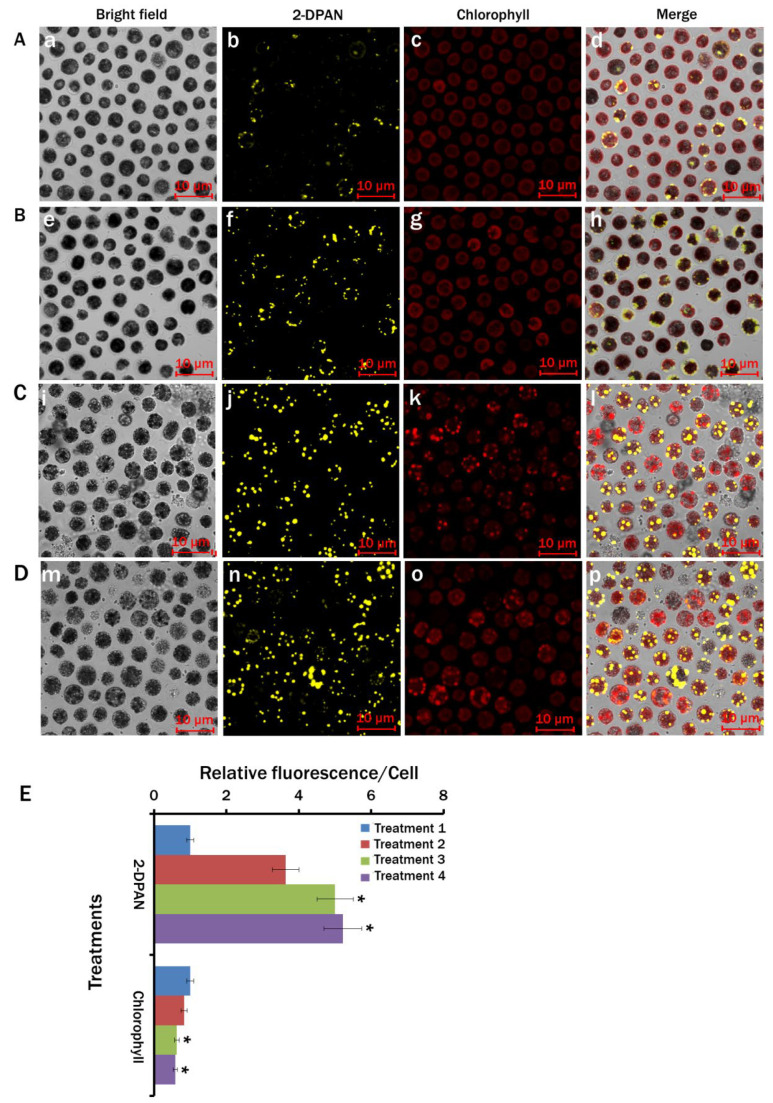
Labelling lipid drops with lipid-specific AIE nanoprobe, 2-DPAN (20 μM; incubation-30 min) in *Chlamydomonas reinhardtii*. Cells were cultured in (**A**) Treatment 1: modified Woods Hole (MBL) medium; (**B**) Treatment 2: MBL, (−) N_2_; (**C**) Treatment 3: MBL, (−) N_2_, (−) Ca^2+^; (**D**) Treatment 4: MBL, (−) N_2_, (−) Ca^2+^, (+) sodium acetate (2.0 g/L) (all the treatments were in 24 h light condition). Bright-field images: Aa,Be,Ci,Dm Fluorescence images-2-DPAN: Ab,Bf,Cj,Dn (λ_ex_: 488 nm, λ_em_: 570–650 nm),and Chlorophyll: Ac,Bg,Ck,Do (λ_ex_: 488 nm, λ_em_: 685–758 nm); Merged images: Ad,Bn,Cr,Dp. (**E**) Relative fluorescence intensity/cell for different treatments. Values are relative to the control condition (Treatment 1: modified Cramer–Myers medium (MCM)). Averages shown as mean ±SE; * *p* < 0.05; Images were taken with Zeiss LSM 880 Airyscan confocal microscope.

**Figure 6 biosensors-12-00208-f006:**
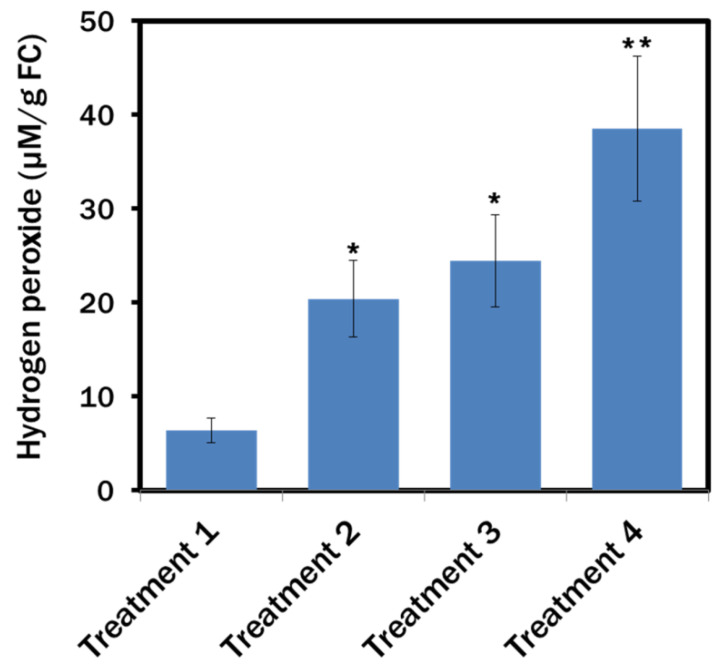
Hydrogen peroxide content in the *Chlamydomonas reinhardtii* cells of different treatments. Treatment 1: modified Woods Hole (MBL) medium; Treatment 2: MBL, (−) N_2_; Treatment 3: MBL, (−) N_2_, (−) Ca^2+^; Treatment 4: MBL, (−) N_2_, (−) Ca^2 +^, (+) sodium acetate (2.0 g/L) (all the treatments were in 24 h light condition). Data represented as mean ± SE, *n* = 3, * *p* < 0.05; ** *p* < 0.01.

**Figure 7 biosensors-12-00208-f007:**
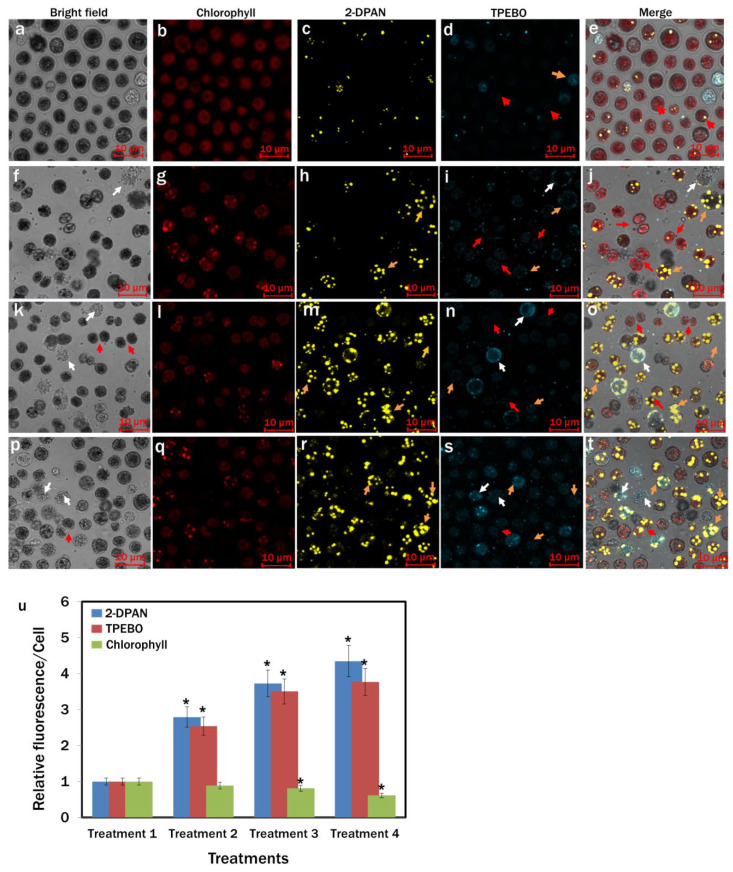
Confocal images of H_2_O_2_ activities in *Chlamydomonas reinhardtii* cells during lipid induction. H_2_O_2_ and lipid drops were labelled with AIE-probes, TPE-BO (100 μM; incubation-15 min) and 2-DPAN (20 μM; incubation-30 min), respectively. (**a**–**e**) Treatment 1: modified Woods Hole (MBL) medium; (**f**–**j**) Treatment 2: MBL, (−) N_2_; (**k**–**o**) Treatment 3: MBL, (−) N_2_, (−) Ca^2+^; (**p**–**t**) Treatment 4: MBL, (−) N_2_, (−) Ca^2 +^, (+) sodium acetate (2.0 g/L) (all the treatments were in 24 hr light condition). Bright-field images: a, f, k and p; Fluorescence images—Chlorophyll: b, g I and q (λ_ex_: 488 nm, λ_em_: 685−758 nm); 2-DPAN: (**c**,**h**,**m**,**r**) (λ_ex_: 488 nm, λ_em_: 570–650 nm); TPE-BO: (**d**,**i**,**n**,**s**) (λ_ex_: 405 nm, λ_em_: 428–499 nm); Merged image: (**e,j**,**o**,) and (**t**,**u**) Relative fluorescence intensity/cell for different treatments. The red arrow indicates H_2_O_2_ activity in normal cells and during cell division; the orange arrow indicates H_2_O_2_ activity in lipid accumulated cells; The white arrow indicates H_2_O_2_ activity in autophagic cells. Averages are shown as mean ±SE; * *p* < 0.05; Images were taken with Zeiss LSM 880 Airyscan confocal microscope.

**Figure 8 biosensors-12-00208-f008:**
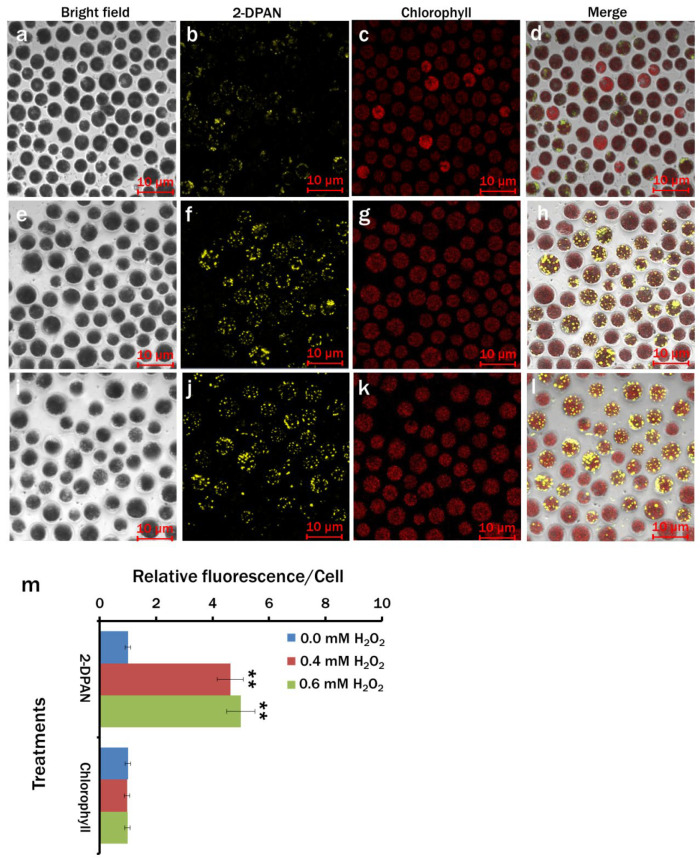
Imaging of lipid drops in H_2_O_2_ treated *Chlamydomonas reinhardtii*. Lipids were labelled with lipid-specific AIE nanoprobe, 2-DPAN (20 μM; incubation-30 min). Cells were cultured in 0.0 mM (**a**–**d**), 0.4 mM (**e**–**h**) and 0.6 mM (**i**–**l**) H_2_O_2_ supplemented MBL medium. Bright-field images: (**a**,**e**,**i**); Fluorescence images—2-DPAN: (**b**,**f**,**j**) (λ_ex_: 488 nm, λ_em_: 570–650 nm) and Chlorophyll: (**c**,**g**,**k**) (λ_ex_: 488 nm, λ_em_: 685–758 nm); Merged images: (**d**,**h**,**l**). (**m**) Relative fluorescence intensity/cell for different treatments. Values are relative to the control condition (0.0 mM H_2_O_2_). Averages shown as mean ± SE; ** *p* < 0.01; Images were taken with Zeiss LSM 880 Airyscan confocal microscope.

**Table 1 biosensors-12-00208-t001:** Percentage of FAMEs in *Chlamydomonas reinhardtii* cells under different cultural conditions.

FAMEs (%)	Treatments					
	Treatment 1	Treatment 2	Treatment 3	Treatment 4	0.4 mM H_2_O_2_	0.6 mM H_2_O_2_
C4:0	1.29 ± 0.18	1.25 ± 0.17	1.01 ± 0.01	0.91 ± 0.15	0.91 ± 0.35	0.72 ± 0.08
C12:0	0.22 ± 0.02	0.21 ± 0.03	0.17 ± 0.01	0.15 ± 0.03	0.15 ± 0.05	0.12 ± 0.03
C13:0	0.16 ±0.042	--*	--*	--*	--*	--*
C14:0	0.87 ± 0.07	0.96 ± 0.05	0.53 ± 0.01	0.25 ± 0.35	0.50 ± 0.06	0.48 ± 0.03
C14:1 N-5	0.12 ± 0.01	0.12 ± 0.01	0.15 ± 0.01	0.11 ± 0.01	0.11 ± 0.02	0.09 ± 0.01
C15:0	0.42 ± 0.03	0.45 ± 0.06	0.26 ± 0.01	0.32 ± 0.03	0.26 ± 0.07	0.23 ± 0.04
C16:0	43.66 ± 1.05	41.10 ± 0.70	37.0 ± 1.28	33.60 ± 0.22	33.60 ± 0.23	33.66 ± 1.0
C16:1 N-7	0.98 ± 0.03	2.01 ± 0.03	1.27 ± 0.22	0.71 ± 0.03	0.56 ± 0.03	0.61 ± 0.16
C17:0	0.95 ± 0.13	0.92 ± 0.13	0.75 ± 0.01	0.71 ± 0.08	0.69 ± 0.23	0.56 ± 0.04
C17:1 N-7	1.79 ± 0.03	2.37 ± 0.08	2.22 ± 0.05	2.19 ± 0.26	1.93 ± 0.23	1.85 ± 0.07
C18:0	2.30 ± 0.03	2.47 ± 0.01	1.70 ± 0.17	1.32 ± 0.19	1.53 ± 0.03	1.50 ± 0.17
C18:1 N-9	5.70 ± 0.78	6.45 ± 1.68	3.06 ± 0.11	1.77 ± 0.23	2.98 ± 0.03	3.19 ± 0.21
C18:2 N-6,9	18.91 ± 0.09	14.13 ± 1.97	12.84 ± 0.60	8.84 ± 0.18	15.75 ± 1.92	16.19 ± 0.28
C18:3 N-3,6,9	17.75 ± 1.53	23.79 ± 2.47	35.19 ± 2.50	46.05 ± 2.31	38.29 ± 2.91	37.38 ± 1.88
C20:0	0.89 ± 0.10	0.87 ± 0.11	0.63 ± 0.01	0.57 ± 0.10	0.59 ± 0.23	0.47 ± 0.03
C20:1 N-9	0.41 ± 0.03	0.39 ± 0.04	0.31 ± 0.01	0.15 ± 0.22	0.27 ± 0.09	0.23 ± 0.02
C20:2 N-6,9	0.42 ± 0.08	--*	0.16 ± 0.23	--*	0.10 ± 0.15	0.23 ± 0.02
C20:3 N-6,9,12	--*	--*	--*	0.17 ± 0.240	--*	--*
C21:0	0.33 ± 0.04	0.32 ± 0.05	0.12 ± 0.18	0.10 ± 0.14	0.24 ± 0.10	0.18 ± 0.02
C20:4 N-6,9,12,15	0.48 ± 0.10	--*	--*	--*	--*	--*
C20:3 N-3,6,9	--*	--*	0.35 ± 0.01	--*	--*	--*
C20:5 N-3,6,9,12,15	0.21 ± 0.30	0.15 ± 0.21	0.15 ± 0.21	--*	0.10 ± 0.14	0.10 ± 0.15
C22:0	0.42 ± 0.35	0.41 ± 0.03	0.33 ± 0.01	0.29 ± 0.05	0.26 ± 0.09	0.22 ± 0.01
C22:1 N-9	1.39 ± 0.17	1.43 ± 0.02	1.55 ± 0.25	1.53 ± 0.69	1.03 ± 0.04	1.79 ± 0.50
C22:2 N-6,9	0.12 ± 0.14	0.05 ± 0.07	0.10 ± 0.01	0.05 ± 0.08	0.03 ± 0.05	0.08 ± 0.01
C22:4 N-6,9,12,15	--*	--*	--*	0.06 ± 0.08	--*	--*
C24:0	0.08 ± 0.01	0.09 ± 0.01	0.03 ± 0.01	0.03 ± 0.01	0.03 ± 0.01	0.02 ± 0.01
C22:6 N-3,6,9,12,15,18	0.06 ± 0.01	--*	0.06 ± 0.08	0.04 ± 0.06	0.04 ± 0.06	0.04 ± 0.06
C24:1 N-9	0.04 ± 0.01	0.01 ± 0.01	--*	0.01 ± 0.01	0.01 ± 0.01	0.01 ± 0.02
SAFAs	51.61 ± 12.41	49.07 ± 11.68	42.55 ± 10.55	38.27 ± 9.59	38.78 ± 9.57	38.18 ± 9.61
MUFAs	10.44 ± 1.97	12.80 ± 2.24	8.58 ± 1.15	6.49 ± 0.89	6.90 ± 1.10	7.79 ± 1.19
PUFAs	37.42 ± 8.04	38.13 ± 8.69	48.86 ± 11.92	55.22 ± 11.25	54.32 ± 13.16	54.04 ± 12.92

* none detectable.

## Data Availability

The data supporting the findings of this study are available from the corresponding author upon request.

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
