# Peer review of "Visualising the Emerging Platform of Using Microalgae as a Sustainable Bio-Factory for Healthy Lipid Production through Biocompatible AIE Probes"

_biosensors, 2022, doi:10.3390/bios12040208_

Round 1
Reviewer 1 Report
In this manuscript, Qin and co-workers are demonstrating the applicability of AIE based fluorescent probe for investigating the ability to track LD content in cells under different conditions. The concept of the manuscript is interesting, and it has a potentially useful application. Author have provided a good experimental set up with the application of basic statistics for the validation of their results. I found few minor errors’/deficiencies which I would like to suggest authors to revise prior to the publication of this manuscript.
(1). The written abstract for the manuscript does not highlight the importance of the findings. I would recommend authors to revise the abstract section carefully.
(2). Can authors provide a scheme in the introduction highlighting some of the important previously reported lipid droplet visualizing probes and then in a comparison to highlight what are the specific advantages of this reporting model?
(3). With corresponding to the figure 2, can authors provide absorbance spectra as well?
(4). For the fluorescence microscopy images authors should provide the intensity correlation plot with the distance to show the colocalization. In addition, authors should clearly indicate what the specific excitation wavelength settings for each dye as well as the emission filter settings to confirm these observed images are not artifacts from channel bleeding. In addition, authors must provide Pearson’s overlap coefficient for the colocalization images. Also, clearly indicate incubation times as well as the specific staining concentrations. This information is vital to provide with the microscopy images. Similarly these information should provide for the figure 5 and all other fluorescence microscopy based images.
(5). In figure 3, how did authors calculate the relative fluorescence /cell?
Reviewer 2 Report
This study is to use microalgae to obtain high-value healthy and beneficial lipids, and shows that this study is a fast and simple in vivo tool to detect lipid production in microalgae with carbohydrate cell wall.
The text of this manuscript is well written, and the supporting data are well stated and explained. However, there are a large number of references and statements of supplementary materials in the text (Fig. S1 to Fig. S8), but the data of any supplementary materials are not seen in the submitted articles.
Reviewer 3 Report
The submitted manuscript by AHM Mohsinul Reza et al. reports on various variants of the lipid-inducing conditions in C. reinhardtii. The treatment variations included different ratios and concentrations of N2, Ca-ion, sodium acetate, or H2O2. The authors have introduced a rapid, wash-free, multicolor imaging technique for the visualization of the liquid droplets and hydrogen peroxide in C. reinhardtii using aggregation-induced emission sensors.
The study is well designed, conducted, and described in the manuscript. The conclusions made by the authors are supported by the data. I recommend however reconsidering several points in the manuscript:
- The manuscript is not easy to read due to numerous abbreviations which are not clarified. Please improve.
- Figure 4 needs re-drawing since the font size is too small.
- Please check Figure 6, axis y "Hydrogen peroxxide"
- Table 1: please explain in the Table 1 caption what is the reason to highlight some of the values in a bold font. Please also explain why for some of the % values one sees quite large standard deviations?
- Please check the formatting of the literature 74, 76, 81, 87.
Round 2
Reviewer 3 Report
I agree with the revised version of the manuscript and recommend its acceptance for publication.